# Subsoil particulate organic matter is more responsive to 10 years

## of whole-soil warming than mineral-associated organic matter in

# a temperate forest

Binyan Sun¹, Guido L. B. Wiesenberg¹, Elaine Pegoraro², Margaret S. Torn², Michael W. I.

Schmidt¹, Mike C. Rowley¹

¹Department of Geography, University of Zurich, Zurich, Switzerland

²Earth and Environmental Sciences Area, Lawrence Berkeley National Laboratory,

Berkeley, California, USA

Corresponding to: Binyan Sun (binyan.sun@geo.uzh.ch)

Mike C. Rowley (mike.rowley@geo.uzh.ch)

#### Abstract

Global average temperatures are forecast to increase by 4°C by 2100 under the Intergovernmental Panel on Climate Change SSP5–8.5 scenario. This warming could accelerate soil organic carbon (SOC) mineralization, net loss of soil carbon to the atmosphere, and consequently exacerbate global warming through a positive feedback loop. It is generally assumed that mineral–associated organic matter (MAOM) is less sensitive to warming compared to particulate organic matter (POM), especially in subsoil; yet more empirical data investigating the whole–soil response to warming is still required to test this assumption.

Our study was conducted in a whole–soil field warming experiment in a temperate mixed–conifer forest at Blodgett Forest Research Station, University of California, Berkeley, which had been subjected to 10 years of warming. Soils taken at three depths (10–20, 40–50, and 80–90 cm) were separated into three density fractions, and we then investigated the SOC concentration (elemental analysis) and composition of bulk soil and fractions with diffuse reflectance infrared Fourier transform (DRIFT) spectroscopy. We found that a decade of experimental warming shifted subsoil bulk SOC composition towards lignin and C–H aromatic bonds. warmed plots had significantly lower mass of the POM fractions relative to control plots in the subsoil (80–90 cm), but there was no difference in the topsoil, which could occur if higher decomposition losses of POM were obscured by fresh plant inputs. In contrast, the mass of MAOM and its chemical composition was not different between warmed and control treatments, but did shift along the depth gradient. This study thus supports the assumption that

https://doi.org/10.5194/egusphere-2025-5483 Preprint. Discussion started: 14 November 2025 © Author(s) 2025. CC BY 4.0 License.

- POM is more responsive to warming than MAOM, which was particularly evident in the
- subsoil at Blodgett Forest.

- Keywords: Soil organic carbon, density fractionation, Diffuse Reflectance Infrared
- Fourier Transform spectroscopy, soil warming, decomposition, carbon cycling.

41

46

52

58

61

66

### 1. Introduction

the atmosphere and vegetation combined (Jackson et al., 2017; Scharlemann et al., 2014; Schlesinger, 1990). Under the SSP5-8.5 scenario, global temperatures are projected to increase by 3.3-5.7°C by 2100 (IPCC, 2023). This warming could accelerate the decomposition of soil organic carbon (SOC) and increase soil carbon dioxide (CO<sub>2</sub>) fluxes (Davidson and Janssens, 2006; Schlesinger and Andrews, 2000). Warming-induced soil CO<sub>2</sub> fluxes could further exacerbate global warming, creating a positive feedback cycle (Davidson and Janssens, 2006). Many studies have examined SOC responses to warming and reported variable outcomes, ranging from no net cumulative changes (Giardina et al., 2014; Lu et al., 2013) to large SOC losses observed in more recent field warming experiments (Nottingham et al., 2020; Soong et al., 2021, p.2; Verbrigghe et al., 2022). The contrasting results reflect the differences in experimental approaches as well as the complexity of soil systems, where opposing mechanisms simultaneously influence SOC inputs and decomposition. As a result, bulk soil measurements alone can lack the sensitivity needed to uncover the processes driving SOC responses (Lavallee et al., 2020; Rocci et al., 2021). Instead, investigating how fractions of soil, each with distinct physical and chemical properties, respond to warming could improve our understanding of the key mechanisms that govern SOC dynamics (Georgiou et al., 2022; Lugato et al., 2021). Soil organic carbon is typically operationally-divided by size or density fractionation into two main fractions, particulate organic matter (POM) and mineral-associated organic matter (Lavallee et al., 2020). POM mainly consists of coarser, partially-decomposed fragments of plant biomass (Poirier et al., 2005; Rocci et al., 2021; Swanston et al., 2005). In contrast, carbon in MAOM includes both microbial-derived products such as necromass and metabolites (Lehmann and Kleber, 2015; Sanderman et al., 2014; Sollins et al., 1999) and plant-derived biomolecules (Angst et al., 2021). Due to its strong association with minerals, MAOM is considered less accessible to decomposers or their extra cellular enzymes, and consistently has longer turnover times (Kaiser and Guggenberger, 2000; Lavallee et al., 2020; Lützow et al., 2006). Therefore, it is also often assumed to be more resistant to warmingenhanced decomposition than POM (Georgiou et al., 2024; Williams et al., 2018). Nevertheless, there is limited empirical evidence to support or refute this assumption (Guan et al., 2018; Schnecker et al., 2016, p.201) and few studies have investigated how the distribution

Soil is the largest actively-cycling terrestrial carbon reservoir, storing more carbon than

of SOC amongst these fractions changes with warming (Chen et al., 2023; Schnecker et al., 2016; Soong et al., 2021).

Warming can not only influence SOC stocks and distribution among fractions, but also carbon and microbial community composition (vandenEnden et al., 2021). Long—term warming can rapidly deplete chemically less complex substrates and shift microbial communities towards decomposers of chemically complex carbon (Melillo et al., 2002; Frey et al., 2013; Stuble et al., 2019). Decomposition of these chemically complex substrates require multiple enzymatic steps for depolymerization (Steinweg et al., 2013), which tend to reduce microbial carbon use efficiency. Lower CUE means a greater fraction of carbon is respired rather than assimilated, potentially reducing microbial necromass formation and its stabilization in soil (Bradford et al., 2016; Crowther et al., 2015; Kögel-Knabner et al., 2008). Furthermore, the composition of SOC can influence its interaction with minerals, as compounds with different functional groups exhibit varying affinities for mineral surfaces (Kleber et al., 2021; Spohn, 2024). Therefore, understanding the composition of SOC, especially in different physical fractions, is essential to disentangling the nuanced shifts of SOC stabilization and destabilization processes due to warming.

Whole-soil warming experiments have been instrumental in advancing our understanding of the response of SOC to warming, particularly in subsoils (> 30 cm), which are estimated to contain ~50 % of SOC globally (Jobbágy and Jackson, 2000). Subsoils differ from topsoils in terms of plant inputs, microbial community, and SOC stabilization mechanisms (Fierer et al., 2003; Hicks Pries et al., 2023; Rumpel et al., 2012; Salomé et al., 2010). As a large, long-term carbon reservoir (Harrison et al., 2011; Rumpel and Kögel-Knabner, 2011; Sierra et al., 2024), it is imperative to understand how the whole-soil profile, including the subsoil, will respond to future warming. The Blodgett Forest whole-soil warming experiment has been investigating how the top meter of soil of a temperate mixed-conifer forest responds to +4°C warming. Studies have thus far demonstrated that 4.5 years of +4°C warming at Blodgett Forest promoted decomposition and SOC loss in the subsoil (Hicks Pries et al., 2017; Soong et al., 2021), reduced fine-root biomass, and accelerated decomposition of chemically complex SOC (Offiti et al., 2021; Zosso et al., 2023). Specifically, the SOC loss in the subsoil at Blodgett Forest was driven by a reduction in POM, while MAOM stocks remained largely unaffected (Soong et al., 2021). Given the paucity of fresh plant inputs to deeper horizons, which limits POM inputs (Button et al., 2022; Hicks Pries et al., 2023; Jackson et al., 1996), subsoil POM losses are unlikely to be promptly replenished. If subsoil POM losses persist over decadal timescales, the resulting lower POM availability could limit microbial

processing and in turn, constrain MAOM formation derived from microbial transformation of POM (Heckman et al., 2022; Witzgall et al., 2021). However, it remains unclear how prolonged warming will affect the balance between POM and MAOM at deep—soil warming experiments such as Blodgett Forest and whether these SOC losses in individual fractions persist or stabilize over decadal timescales.

To address this knowledge gap, we investigated responses in the distribution of SOC and its composition across fractions and depths after 10 years of warming at Blodgett Forest. We fractionated soil from soil cores using density fractionation separating the free light particulate organic matter (fPOM), occluded particulate organic matter (oPOM), and MAOM. We quantified bulk soil and fraction carbon content, stable carbon isotope composition ( $\delta^{13}$ C values), and quality using diffuse reflectance infrared Fourier transform (DRIFT) spectroscopy. We hypothesized that under long–term warming, fPOM and MAOM would remain quantitatively stable and be characterized by aliphatic C–H due to replenishment of fresh plant inputs which offsets losses in topsoils. In contrast, fPOM would be depleted and MAOM would remain consistent quantitatively in subsoils under warming, and both fractions become more aromatic.

### 2. Materials and methods

#### 2.1 Study site and sampling

The whole–soil warming experiment is at the Blodgett Forest Research Station, run by the University of California, Berkeley, located in the foothills of the Sierra Nevada near Georgetown, California (38°54'43.0"N 120°39'40.0"W; 1370 m above sea level). The climate is characterized as Mediterranean with a mean annual air temperature of 12.5°C and a mean annual precipitation of 1774 mm, falling predominantly between November through April (Hicks Pries et al., 2017). The experiment is located in a mixed coniferous forest dominated by ponderosa pine (*Pinus ponderosa*), sugar pine (*Pinus lambertiana*), incense cedar (*Calodefrus decurrens*), white fir (*Abies concolor*), and douglas fir (*Pseudotsuga menziesii*; Rasmussen et al., 2005; Hicks Pries et al., 2017). The understory vegetation mainly consists of shrubs *Notholithocarpus densiflorus*, grasses, and annual plants such as *Gallium triflorum*. The soil is a mesic ultic Alfisol of granitic origin, which is equivalent to Dystric Cambisols (IUSS Working Group WRB, 2022), with fine–loamy texture.

The warming experiment started in January 2014 and was described in detail by (Hicks Pries et al., 2017). Briefly, it consists of three pairs of circular plots that are 3 m in diameter,

with vertically installed heating cables down to 1 m. Each pair contains a control plot, with the temperature in warmed plots elevated by +4°C compared to the control plots. The experimental setup between warmed and control plots is identical except that heating was not turned on for control plots. In May 2023, two cores from each plot were collected in 10 cm depth increments down to 1 m depth. Samples were stored at -4°C until processing.

### 2.2 Soil preparation

Samples were sieved to 2 mm to separate the fine-soil fraction (< 2 mm) from large rock fragments and roots. Tweezers were used to manually remove roots that passed the sieve. The soil samples were then freeze-dried to a constant weight. Before laboratory analysis, samples from identical depth increments of the two different cores from the same plot were combined to obtain a more representative bulk sample in an effort to account for within-plot variability. A subset of the samples was then ground with a ball mill (MM400, Retsch, Haan, Germany) for elemental analysis.

Unless otherwise specified, topsoil and subsoil will refer to soil between 0–30 cm and 30–100 cm, respectively. Surface soil and deep soil will refer to the soil depth of 10–20 and 80–90 cm throughout the Results and Discussion sections.

#### 2.3 Density fractionation

Bulk soil samples from three depths (10–20, 40–50, and 80–90 cm) were fractionated by density into free POM (fPOM, large undecomposed or partially decomposed plant fragments), occluded POM (oPOM, plant fragments released by sonication), and MAOM (residual SOC bound to minerals) fractions in sodium polytungstate solution (SPT), with methods adapted from Hicks Pries et al. (2018). Like Hicks Pries et al. (2018) and (Soong et al., 2021) we used a density of 1.65 g cm<sup>-3</sup>. The three depths were selected in particular as they represented different soil horizons and allow for comparison with previous analyses at the same experiment site (Hicks Pries et al., 2017; Soong et al., 2021).

Unlike (Hicks Pries et al. (2017) and Soong et al. (2021), who used 20 g bulk soil for density fractionation, we conducted fractionation in 80 mL round bottom glass centrifuge tubes (Neubert–Glas, Germany) to avoid potential plasticizer contamination for the following lipid analysis. Briefly, we added 40 mL of 1.65 g cm<sup>-3</sup> low C/N SPT (SPT0, TC–Tungsten Compounds Inc., Grub am Forst, Germany) to 8 g bulk soil in a round bottom glass centrifuge tube with four replicates per sample. The glass tube was gently shaken by hand to ensure full contact between the soil particles and the solution, and any particles adhering to the tube walls

185

186

187188

were brought back into the solution. We let the samples stand for 1 h to allow for density 167 equilibration and maximize the separation of fPOM from MAOM before centrifuging the 168 solutions in a swinging bucket rotor for 1 h at 3130 g (Megafuge 1.0, Heraeus Group, 169 Germany). The fPOM was aspirated using a 10 mL Eppendorf pipette, filtered through a 0.8 170 μm polycarbonate filter (Nucleopore Track-Etch, Whatman), and rinsed with deionized water (Millipore MilliQ Advantage A10, Darmstadt, Germany, 18.2 MΩ.cm at 25 °C, 2 ppb TOC). 171 172 The remaining sample was then sonicated in an ice bath at a maximum amplitude and 50 % 173 pulse for 2 mins and 26 seconds, delivering a calibrated (North, 1976) total energy input of 100 174 J mL<sup>-1</sup> (Hicks Pries et al., 2018) The sample was then centrifuged, again at 3130 g for 1 h and 175 left to settle overnight. Subsequently, the same aspiration and rinsing procedure used for fPOM 176 was applied to the floating material, fractionating the oPOM. To prevent the re-adsorption of 177 oPOM onto the MAOM and to remove all the oPOM, the remaining fraction was centrifuged 178 (3130 g for 1 h) aspirated, filtered, and rinsed a second time. All fractions, including the 179 remaining MAOM were then rinsed with deionized water until the supernatant reached the 180 density of water. All fractions were freeze-dried at -50°C (Alpha 1-4, Martin Christ Freeze 181 Dryers, Osterode am Harz, Germany). Then a subsample of fPOM and oPOM were ground by 182 hand with a pestle and agate mortar and a subsample of MAOM was ground with a ball mill 183 (MM400, Retsch, Haan, Germany) for further elemental and DRIFT analysis.

### 2.4 Carbon and nitrogen concentrations and carbon stable isotope compositions

Soil organic carbon concentrations (%C) and stable carbon isotope compositions ( $\delta^{13}$ C values) were analyzed in bulk soils and fractions using an elemental analyzer–isotope ratio mass spectrometer (EA–IRMS; Flash 2000–HT Plus, linked by Conflo IV to Delta V plus isotope ratio mass spectrometer, Thermo Fisher Scientific, Bremen, Germany). Caffeine (Merck, Germany) and Chernozem (Harsum, Germany) were used as calibration materials.  $\delta^{13}$ C values were expressed in ‰ relative to the V–PDB standard (Conant et al., 2011). At least two analytical replicates were measured for all samples. Carbonates are absent at Blodgett Forest, so total carbon concentrations are assumed to represent organic carbon concentrations.

### 2.5 SOC composition analysis

To assess potential differences in SOC composition caused by warming across soil depths in both bulk soil and fractions, ground soil and fraction samples were analyzed using DRIFT spectroscopy. A Bruker Invenio R spectrometer (Billerica, Massachusetts, USA) was used to record the mid–infrared spectra with a resolution of 2 cm<sup>-1</sup> between 4000 to 80 cm<sup>-1</sup>,

206207

collecting 64 scans for each sample. Background reflectance was determined on oven-dried KBr (60°C) and subtracted to convert spectra to pseudo-absorption units (log [1/R]). Background-corrected scans were checked for consistency and then averaged to produce a single spectrum.

All subsequent spectral processing was completed in R version 4.4.2 (R Core Team, 2025) using the 'prospectr' (v0.2.8; Stevens and Ramirez-Lopez, 2025) and 'tidyverse' (v1.3.2; Wickham et al., 2019) packages. Briefly, spectra were concatenated into a single datafile and trimmed to between 4000-400 cm<sup>-1</sup> to remove spectral noise. Prior to standard normal variate normalization (SNV), the data was smoothed using a Savitzky-Golay filter to the 3<sup>rd</sup> order polynomial (Savitzky and Golay, 1964) and scaled to positive values (Fearn, 2008). SNV was applied to the spectra to eliminate scattering effects and standardize spectral intensities, before applying a convex-hull rubber-band baseline correction. Area under the curve (AUC) of spectral peaks was calculated using a trapezoid area function with local baseline calculation, to ascertain the area of peaks linked to different bonding environments of SOC. The spectral band ranges assigned to different carbon functional groups were based on previous studies (Artz et al., 2008; Chatterjee et al., 2012; Ofiti et al., 2021) and are reported in Table S13. The average AUC plot for each soil sample type (bulk, fPOM, oPOM, MAOM) was calculated and used to minimally adjust the wavenumber range of spectral bands for each peak due to slight spectral shifts caused by the mineral matrix, detailed in Table S13 (Ellerbrock and Gerke, 2021).

The ratio of peak areas between aromatic C=C/carboxylic C=O stretches (in our paper assigned as C=C aromatic2; Table S13) and aliphatic C-H stretches were used to calculate the DRIFT Stability Index (Demyan et al., 2012; Haberhauer et al., 1998; Laub et al., 2020; Schiedung, 2025), indicative of the relative state of SOC decomposition. In soils devoid of carbonates, the band range at ~3000–2800 cm<sup>-1</sup> is largely unaffected by mineral–phase interference in the absorbance (Tinti et al., 2015). Other disturbances to the spectra should be minimized because we always compare samples collected from close proximity with a similar parent material and texture (Reeves, 2012). We could not conduct DRIFT analysis on several POM fractions from the deep soil due to the limited amount of material recovered, particularly in warmed fractions.

### 2.5 Statistical analysis

All statistical analyses were performed in R version 4.4.2 (R Core Team, 2025) using the Rstudio (2024.12.1.563; Posit team, 2025). Plots were all created using the 'ggplot2'

package (v3.5.1; Wickham, 2016) and 'ggbiplot' (v0.6.2; Vu et al., 2024). To test the effect of depth, treatment, and their interaction on bulk soil carbon concentration and  $\delta^{13}$ C values, and the DSI, we built linear mixed effects models (LMEs) in the 'nlme' package (v3.1.166; Pinheiro et al., 2000; Pinheiro et al., 2025). We set depth, treatment, and their interaction as fixed effects, setting the treatment block as a random effect. Model fits were assessed using Akaike's Information Criterion (Akaike, 1998). Homoscedasticity of residuals were visually examined using conditional residual plots (Zuur et al., 2009) and normality was assessed using Q–Q plots. If model assumptions were not met, as was the case with SOC concentration, the data were log-transformed. To account for autocorrelation of observations within profiles, depth class was set as a repeated measure with an autoregressive (AR1; 'nlme') covariance structure (Grand et al., 2014).

For fractions, we applied LMEs to the three fractions separately. We tested SOC concentration in each fraction (mg OC  $g^{-1}$  fractionated soil), distribution of SOC in the density fractions (g C  $g^{-1}$  SOC), and DSI, in response to treatment, depth, and their interaction as fixed effects, and block again as random effect. When significant fixed–interaction effects were discovered, we ran post hoc analyses within each depth increment. The alpha level was set to  $\alpha = 0.05$  in all statistical tests; a p 

273274

275276

and 56 % (CI: -84 %, 20 %; p = 0.086; Table S12), respectively. Independently of treatment,

SOC concentration significantly declined with depth (p 

Fig. 1. Soil organic carbon (SOC) concentration in each soil fraction (mg OC  $g^{-1}$  fractionated bulk soil) at three depths (10–20, 40–50, and 80–90 cm) in control and warmed plots (n = 3), error bars represent the standard error of the mean. The asterisks indicate the significant treatment effects (p < 0.05 '\*'; < 0.01 '\*\*') on the mass of SOC in the corresponding recovered fraction.

We did not find significant effects of interaction between soil depth and warming on the fPOM proportion to total SOC (g C g<sup>-1</sup> bulk SOC; Table S5). The fPOM proportion significantly decreased with depth (p = 0.032; Table S5) and on average ranged from 49 % to 11 % in warmed plots and from 41 % to 18 % in control plots across soil depth (Fig. 2). The oPOM proportions were consistent without being significantly affected by either interaction between soil depth and warming, and main effects (Table S5). oPOM proportions remained stable across soil depth (Fig. 2). Warming consistently but non–significantly reduced oPOM proportion at individual depths (Fig. 2), in particular at 80–90 cm, where the decline of oPOM proportion was marginally significant (p = 0.057; Table S6). There were no effects of interaction between soil depth and warming on MAOM proportion (Table S5) but its proportion significantly increased with depth (p = 0.032; Table S5). On average, the proportion of MAOM increased from 41 % to 66 % in control plots and from 38 % to 81 % in warmed plots across soil profile (Fig. 2).

Fig. 2. Distribution of total soil organic carbon (SOC) in the density fractions (g C  $g^{-1}$  total SOC) at three depths (10–20, 40–50, and 80–90 cm) in control and warmed plots (mean  $\pm$  SE, n = 3).

### 3.3 Bulk SOC composition change

Bulk SOC composition changes between warmed and control plots are displayed in the PCA biplot (Fig. 3). The first two principal components explained 44.9 % (1st) and 19.4 % (2nd) of total variation in the bulk DRIFT dataset (Fig. 3). SOC composition changed significantly with depth moving across principal component 1; the topsoil samples (0–30 cm) were clearly separated from subsoil samples (> 30 cm). Soil above 30 cm did not cluster according to treatment and was associated with an increased AUC values for aliphatic and C=C aromatic (range1) regions of the DRIFT spectra (Fig. 3). Samples from 30–40 cm depth contained higher proportions of polysaccharide and carboxylic functional groups and showed no clear treatment—based separation. In contrast, spectra from subsoils > 40 cm were primarily grouped by treatment in the PCA plot and separated along principal component 2. Control subsoils were characterized by an increased presence of C=C aromatic (range 2) and C-H aromatic (range 2; Fig. 3); while warmed subsoils 

Fig. 3. Principal component analysis (PCA) of the area under the curve values from fit Diffuse Reflectance Infrared Fourier Transform spectra of bulk soil in control and warmed plots at 10 cm intervals from 0 to 100 cm. Each point represents the mean PCA scores of the three replicate samples per depth and treatment. The horizontal (PC1) and vertical lines (PC2) indicate the standard error of the mean of principal component scores for each plot type (control and warmed, n = 3), respectively. The number after aromatic C=C/aromatic C-H refers to the same C-bond type assigned at different band ranges (Table S1).

### 3.4 SOC compositional changes in soil density fractions

The PCA of POM fractions is displayed in the supplementary information (Fig. S2). Briefly, the main differences in POM composition were observed between the free and occluded POM fractions rather than under warming or depth. Although SOC in both fractions was composed primarily of aliphatic SOC (Table S7; Fig. S5; Fig. S6), oPOM contained relatively more aliphatic carbon than fPOM. Neither warming nor depth significantly altered the SOC composition within each fraction (Fig. S2). However, warming tended to increase the variability of POM composition, as indicated by the broader dispersion of the standard errors of the mean of warmed samples in the PCA plot.

The first two principal components explained 48.6 % (1<sup>st</sup>) and 30.6 % (2<sup>nd</sup>) of the total variation in the fit DRIFT data of the MAOM fraction (Fig. 3). SOC composition in MAOM

exhibited a marginal depth–related trend along principal component 2, with MAOM at 10–20 cm relatively enriched in aliphatic and C=C aromatic (range 1) bonds, while those from 40–50 and 80–90 cm were more associated with lignin and C–H aromatic (range 2) bonds. Warming did not alter MAOM composition at individual depths, but substantially increased the variability in SOC composition at 80–90 cm.

**Fig. 4.** principal component analysis (PCA) of area under the curve values from fit Diffuse Reflectance Infrared Fourier Transform spectra of MAOM from control and warmed plots at three depth intervals (10–20, 40–50, 80–90 cm). Each point represents the MAOM of one sample. The number behind aromatic C=C/aromatic C-H refers to the same carbon bond type assigned at different band ranges (Table S1).

### 3.5 Changes of aromatic/aliphatic ratio in bulk soil and fractions

We did not find significant effects of interaction between soil depth and warming but significant effects of depth (p 

For fPOM and oPOM, we did not observe either significant interaction effects between soil depth and warming, and main effects (Table S11). On average, warming non–significantly decreased fPOM DSI from  $0.29 \pm 0.07$  to  $0.23 \pm 0.06$  at 10-20 cm and from  $0.27 \pm 0.02$  to  $0.18 \pm 0.03$  at 40-50 cm. Simultaneously, warming non–significantly decreased oPOM DSI from  $0.18 \pm 0.03$  to  $0.14 \pm 0.08$  and from  $0.16 \pm 0.01$  to  $0.11 \pm 0.02$  at 10-20 and 40-50 cm, respectively (Table S10). Although there were no significant effects of interaction between soil depth and warming on MAOM DSI, it significantly increased with depth (p 

5% (CI: -69 %, 190 %) at 40–50 cm, whereas it increased MAOM DSI by 71 % (CI: -44 %, 422 %) at depth, but these effects were weak.

Fig. 5. The Diffuse Reflectance Infrared Fourier Transform (DRIFT) spectral stability index (DSI) or ratio between area under the curve values of C=C aromatic (range 2) to aliphatic in bulk soil (A) at 10 depth increments from 0 to 100 cm and MAOM (B) at three soil depths (10–20, 40–50, and 80–90 cm) under control and warmed conditions (mean  $\pm$  SE, n = 3). The ratio (C=C aromatic2/aliphatic) was calculated using the area under the curve values corresponding to the aliphatic (3020–2800 cm $^{-1}$ ) and C=C aromatic (range 2; 1670–1600 cm $^{-1}$ ) regions. The ratios of POM fractions are shown in Table S10.

### 4. Discussion

### 4.1 Depth-specific shifts in SOC quantity and composition under long-term warming

### 4.1.1 Subsoil carbon loss to warming

Ten years of whole-soil experimental warming caused significantly depth-dependent changes in SOC concentration, consistent with previous observations from the site after 2 and

4.5 years of warming (Hicks Pries et al., 2017; Soong et al., 2021). Notably, soils above and below 50 cm responded in opposite directions. 80 % of fine root biomass appeared in the top 30 cm soil at Blodgett (Hicks Pries et al., 2017), warming likely stimulated root growth into deeper soil horizons and enhanced plant inputs in the soil above 50 cm (Jackson et al., 1996; Kwatcho Kengdo et al., 2025; Leppälammi-Kujansuu et al., 2014). This new input was evidenced by on average non–significant higher SOC concentration, bulk soil DSI, and lower DSI in fPOM fraction. The enhanced inputs offset soil carbon losses due to warming–induced accelerated microbial decomposition (van Gestel et al., 2018; Wang et al., 2025).

Warming consistently reduced SOC concentration in subsoil below 50 cm, with the strongest declines at 60–70, and 80–90 cm, indicating subsoil carbon loss. This agrees with previous results at this site (Hicks Pries et al., 2017; Ofiti et al., 2021; Soong et al., 2021) but contrasts findings from various field–warming studies (Jia et al., 2019; Khan et al., 2025; McGrath et al., 2022; Pegoraro et al., 2021; Wilson et al., 2016), underscoring context– and ecosystem–dependent subsoil responses (Hicks Pries et al., 2023). The observed subsoil loss appears to be driven largely by marginally significant declines in fPOM and significant declines in oPOM. Given limited plant inputs at depth, losses of the relatively available fPOM are insufficiently offset by new carbon supply. Therefore, whether this carbon loss in subsoil will persist or be further amplified over decades remains uncertain, because substrate depletion can temper warming–induced increases in SOC decomposition (Romero-Olivares et al., 2017).

#### 4.1.2 Shifts in subsoils carbon composition

Soil carbon composition displayed marked changes with depth, with a decrease in aliphatic C–H bonds and increase in aromatic bonds, consistent with numerous studies (Chen et al., 2018; Kögel-Knabner, 2002; Rumpel et al., 2002; Schöning and Kögel-Knabner, 2006). Yet, warming–induced changes in bulk SOC composition were depth–dependent. Although we did not observe SOC composition shifts above 40 cm, other studies report contrasting results that warming enhances microbial activity, accelerating decomposition of labile, lignin–, and cuticle–derived carbon in a mix–temperate forest (Feng et al., 2008) or hardwood forest (Pisani et al., 2015; vandenEnden et al., 2021). Similar to our study, (Schnecker et al., 2016) found that 7 years of warming did not alter soil carbon composition in topsoils, likely due to balanced effects of increased microbial degradation and root litter inputs. Although (Ofiti et al., 2021) reported decreased fine and coarse root mass primarily in subsoils > 30 under warming cm at our site, warming–driven new inputs extending to 50 cm could have obscured detectable shifts in SOC composition after decadal warming.

Subsoil SOC compositional shifts were consistent with prior results at this site (Ofiti et al., 2021). Because C=C aromatic (range 2) was the second most abundant bond in fPOM fraction across depths and treatments (Table S8), the depletion of fPOM fraction may have weakened its signal at bulk level. Therefore, a marginal significant loss of fPOM fraction at depth with warming likely drove the on average non–significant decrease of DSI (Fig. 5A) and the shift from C–H/C=C aromatic (range 2) in control subsoils to lignin and C–H aromatic (range 1) in warmed subsoil. With MAOM dominating the warmed subsoil (Fig. 2), bulk signals predominantly reflect mineral–associated bond signals, implying that lignin–mineral association contributes to carbon preservation at depth under warming. Collectively, warmed bulk subsoils exhibited a more advanced decomposition stage.

### 4.2 Divergent roles of different density fractions for carbon sequestration

#### 4.2.1 fPOM as a dynamic pool of soil carbon

The fPOM fraction responded pronouncedly to warming at 80–90 cm depth, which was also observed by (Soong et al., 2021). It is typically assumed that fPOM should exhibit the strongest response to warming, compared to other density fractions, due to its high microbial accessibility and low degree of physical protection (Heckman et al., 2022; Lavallee et al., 2020; Rocci et al., 2021). Yet the inconsistent results from field warming studies, with no fPOM loss (Chen et al., 2023; Schnecker et al., 2016; Soong et al., 2021), or reduced fPOM (Song et al., 2012) in topsoils demonstrated that the responses of fPOM fraction to warming are not straightforward.

One explanation for this discrepancy is the influence of warming on plant inputs, which are significantly correlated with fPOM (Song et al., 2012). When warming increases plant productivity (Ruehr et al., 2023), the warming–induced loss of fPOM fraction in surface soil could be offset (Liu et al., 2025). Moreover, low quality litter from mixed–coniferous forests (Silver and Miya, 2001) such as at Blodgett Forest could limit microbial breakdown (Lavallee et al., 2020) when warming augments inputs, causing fPOM fraction accrual (Crow et al., 2009). (Heckman et al., 2022) suggested that warming effects on decomposition are attenuated by depth due to substrate limitation (Ahrens et al., 2020) and thus lower microbial activity in deep soils. Yet we see loss of fPOM mass at depth at our site, which was previously linked to enhanced microbial activity (Ofiti et al., 2021; Soong et al., 2021) and increased relative abundance of Actinobacteria (Zosso et al., 2021). This group of bacteria is known to utilize more complex carbon (Barret et al., 2011; Goodfellow and Williams, 1983). Therefore, we

hypothesize that the combination of increased microbial decomposition (Soong et al., 2021) and absence of fresh plant inputs jointly contributed to the observed fPOM decline in subsoils.

Consistent with changes in fPOM concentrations, warming did not significantly alter fPOM composition at 10–20 and 40–50 cm (Fig. S1). An overall lower DSI in warmed fPOM suggests a greater contribution from fresh plant inputs compared to the control plots. These inputs likely replenished fPOM despite faster decomposition, maintaining its composition consistent even as turnover accelerated. At 80–90 cm, very low fPOM recovery in warmed plots precluded compositional analysis. The absence of fPOM in the warmed subsoil further supports the hypothesis that there was a disproportionate loss of this fraction under warming. Taken together, our results indicate that fPOM is sensitive to warming, especially in deep soil where enhanced decomposition is not compensated by increased plant inputs.

### 4.2.2 oPOM is an equally dynamic, but a small pool at Blodgett

The oPOM fractions exhibited a similar depth–specific response to warming as fPOM (Fig. 1). Soong et al. (2021) found no significant reduction in oPOM throughout the soil profile, including the 80–90 cm interval. These findings suggest that the prolonged warming might have declined the aggregate turnover time of occluded SOC in the subsoil through time, supporting observations in several other studies (Chen et al., 2023; Poeplau et al., 2020). However, in contrast to fPOM, oPOM represented the smallest density fraction by mass and proportionally in our dataset, which was consistent with previous observations from Blodgett (Hicks Pries et al., 2017, p.17; Soong et al., 2021) and other studies (Schnecker et al., 2016; Schrumpf et al., 2013), and suggested that changes to this pool are quantitatively less significant to bulk SOC.

The SOC composition of oPOM was highly heterogeneous and showed no clear trends with depth or warming. This is evident in the wide spread of oPOM compositions along PC1 and PC2 (Fig. S2), reflecting the heterogeneity of SOC preserved by aggregates (Wagai et al., 2009; Wiesenberg et al., 2010). This heterogeneity is seen in other studies where oPOM is often associated with a broader range in  $\delta^{13}$ C and  $\Delta^{14}$ C values (Marín-Spiotta et al., 2008; McFarlane et al., 2013; Rowley et al., 2021; Schrumpf et al., 2013). The reason for this is the heterogenous nature of this fraction containing organic matter at various stages of decomposition, which, if not further separated, can result in strongly variable chemical compositions in space and time (Dorodnikov et al., 2011; Wiesenberg et al., 2010). However, as the mass of the oPOM fraction was very limited, further separations followed by chemical analyses were not feasible in the current study. In summary, oPOM sensitivity to warming may

have increased with time to reflect that of the fPOM, but it was a quantitatively small pool of SOC at Blodgett Forest.

### 4.2.3 Subsoil MAOM as a long-term carbon preservation pool

The mineral–associated organic matter was consistently less responsive to warming than fPOM and oPOM across the soil profile (Figs. 1–2). This provides among the first fraction–resolved evidence that subsoil MAOM remained consistent in both mass and chemical composition under decadal warming.

Direct sorption and microbial assimilation are the major drivers of MAOM formation in topsoils (Mikutta et al., 2019). Sufficient fPOM supplies sufficient substrates that can be sorbed to minerals via *in vivo* pathway (Cotrufo et al., 2013; Liang et al., 2017; Witzgall et al., 2021). In our warmed topsoils, greater plant inputs (Castellano et al., 2015) with elevated microbial activity likely offset MAOM loss (Cotrufo et al., 2013; Islam et al., 2022), resulting in constant MAOM. This pattern aligns with other warming studies (Schindlbacher et al., 2009; Schnecker et al., 2016) but contrasts with (Chen et al., 2023), who documented warming—induced decreased inputs and suppressed decomposition leading to a significant MAOM loss. Given the continuous exchange of MAOM and bulk SOC in topsoil, reflected by similar DSI of both, we infer that warming accelerated turnover and shortened MAOM residence time (Mikutta et al., 2019), especially relative to subsoil MAOM. In our subsoils, MAOM remains stable likely because accelerated fPOM and oPOM decomposition provided substrates that can be adsorbed to minerals. Another possible reason would be that warming can also enhance vertical transport of DOC (Soong et al., 2021), which can be efficiently associated with minerals (Kramer et al., 2012; Mikutta et al., 2019; Villarino et al., 2021; Yu et al., 2022).

MAOM composition remained consistent in subsoils under warming, characterized mainly by both C=C aromatic (range 2) and lignin signals (Table S8), which did not hold at bulk level (Fig. 3). This affirmed the substantial loss of C=C aromatic (range 2) from fPOM under warming, which aligns with significant loss of aromatic—binding containing polymers documented by (Zosso et al., 2023) at this site, although these bonding types represent chemically complex that typically accumulate during microbial decomposition (Calderón et al., 2013; Demyan et al., 2012; Haberhauer et al., 1998). Its decline points to mineral protection rather than intrinsic recalcitrance as the key process for SOC preservation (Marschner et al., 2008). Such organo—mineral associations may account for the pronounced lignin signal and higher average DSI in warmed deep soil, because phenolic—rich lignin derivatives are readily stabilized by Fe/Al oxides and to some extent Ca (Kögel-Knabner et al., 2008; Rowley et al.,

2024). Concurrently, a consistent decrease in aliphatic signals in MAOM in deep soil under warming was also observed, indicating that the strength of organo—mineral associations may shape its warming responses (Conant et al., 2011; Kaiser et al., 1996; Lützow et al., 2006). However, resolving the underlying mechanisms was beyond the scope of the current study and warrants targeted future work. Collectively, the small changes in its concentration and composition support the notion that MAOM is a slow—cycling carbon pool at Blodgett, which buffers soil carbon loss under decadal whole—soil warming and can be regarded as a carbon reservoir to help mitigate climate change.

### 5. Conclusion

Our study is the first to investigate the SOC composition across different soil fractions within a whole–soil field warming experiment after 10 years of warming. Our findings highlight that bulk soil and SOC fractions responded differently to long–term warming, with fPOM and oPOM being more responsive than MAOM over a decadal timescale. The loss of dynamic fPOM fraction drives the consistent and marginally significant decrease of SOC concentration at certain depths in subsoil. This has direct implications for carbon–climate feedback modeling, as initial warming–enhanced CO<sub>2</sub> fluxes (Hicks Pries et al., 2017; Soong et al., 2021) may be attenuated over decadal timescales by the depletion of this labile POM, in particular in subsoil. MAOM showed much lower responses to warming than the POM fraction, and aromatic compounds in association with minerals is a key mechanism for SOC preservation at Blodgett. We demonstrate that combining DRIFT spectroscopy with density fractionation provides a more powerful approach for better understanding of SOC dynamics under warming than analyzing bulk soil alone. Therefore, future long–term whole–soil warming experiments should place particular emphasis on subsoils and individual SOC fractions to better understand depth–specific carbon dynamics.

### Competing interests

The authors declare that they have no conflict of interest.

### Code/Data availability

#### **Author contributions**

BS conducted all the lab work, contributed to field work, wrote the original draft, conducted statistical analysis and created the figures. GLBW supervised BS, contributed to methods, conceptualization, data interpretation and validation, edited, and reviewed the manuscript. MWIS obtained funding for Zurich costs, and contributed to conceptualization, data interpretation, writing review, and editing. EP led the field work, and contributed to statistical analysis, writing review. MST designed and is PI of the Blodgett whole—soil warming experiment, contributed to field work and edited the manuscript. MCR developed the DRIFT spectra analysis pipeline for spectral analysis and calculation, contributed to statistical analysis, conceptualization, data interpretation and validation, edited and reviewed the manuscript.

### Acknowledgments

This study was supported by the Swiss National Science Foundation (SNF) funded DEEP C project (200021\_172744) and the Belowground Biogeochemistry Scientific Focus Area funded by the U.S. Department of Energy, Office of Science, Office of Biological and Environmental Research, Environmental System Science Program, under Contract Number DE–AC02–05CH11231.We thank Jan Pfiffner, Julia van Leeuwen, Thomas Keller, Barbara Siegfried and Yves Brügger for support with laboratory analyses. We thank Cristina Castanha and Niklas Blanadet for support with field work and Rachel C. Porras for training on density fractionation.

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
