# Peer review of "Subsoil particulate organic matter is more responsive to 10 years"

_EGUsphere, 2025_

## Author Comment (AC1)

**RC1**

Dear editor and authors, thank you for your invitation and sorry for the delay comments.

This manuscript presents an interesting investigation of deep soil carbon stability under warming conditions, based on the well-established field experiment at Blodgett Forest. The study is valuable, and the experimental design and methodological approach are generally sound. However, several aspects require substantial improvement before the manuscript can be considered further.

In particular, data quality should be carefully re-evaluated, as large standard errors are observed for several parameters. In addition, the figures and tables need to be reorganized, and statistical analysis results should be clearly indicated or labeled in figures and tables. The English language requires further polishing, and the citation format in the main text should be carefully checked and revised, as the current format occasionally impedes readability and comprehension.

Please see my detailed comments below.

Thanks a lot for your careful reading, insightful comments, and constructive suggestions. We really appreciate your time, and thorough review of the manuscript, which has greatly increased its quality. We will address your comments below accordingly.

**General Comments**
**Introduction**

The Introduction would benefit from additional background on SOC functional groups, as these constitute one of the main parameters measured in this study. Moreover, free light POM and occluded POM are not sufficiently introduced and should be described more clearly.

Thank you for your comment, we will add further definitions and exploration in the introduction. Specifically, we will add:

1. A background on SOC functional groups.
2. A definition of occluded POM (oPOM).
3. Differences between fPOM and oPOM.

**Results**

This section is generally difficult to follow. First, many results are described primarily based on tables and figures presented in the Supplementary Material rather than in the main text. The authors should consider integrating key results into the main text or consolidating them into clearer, more informative figures.

Second, numerous non-significant results are reported, which is not necessary and detracts from the main findings.

In addition, I strongly recommend conducting ANOVA followed by appropriate post hoc tests for parameters across different soil depths and indicating statistically significant differences using uppercase or lowercase letters in figures and tables.

Thanks a lot for your critical, constructive comments on our results section. We will address your comments as below:

1. We have added the SOC concentration data and PCA plot for fPOM and oPOM data to the main paper as a table/figure.

2. Second and third points will be replied together since they are all related to statistics. We used linear mixed effects models (LMEs) rather than ANOVA test because: 1) the same statistical methods have been used in previous studies (Ofiti et al., 2021; Soong et al., 2021; Zosso, 2022) and we wanted to remain consistent; 2) ANOVA does not account for random effects such as block-based design in Blodgett Forest, does not fit for unbalanced data (for fPOM and oPOM), and most importantly does not account for autocorrelation (or pseudo-replication) due to the assumption of independence. The vertically adjacent depths are not independent from each other, which will violate the basic assumptions of ANOVA (independent observations). Our LMEs use a covariance structure (auto-regressive 1) to account for this lack of independence. However, thank you for your suggestion.

In general, we prefer conservative statistical analysis as a supportive tool for ecological, and biological interpretation of our data. Therefore, together with small sample size and high spatial heterogeneity, we also report p values that are between 0.1 and 0.05 as marginally significant. Also, as we mentioned in the M&M, we include interaction in the LMEs, and only run post hoc tests when the interaction was significant. We will remove

supplementary tables after reporting p-values to avoid potential confusion. One can also check the original data on ESS-dive website once this paper is accepted/published.

**Discussion**

The Discussion requires further improvement. The authors should focus more strongly on their own results and novel findings, rather than extensively mixing their interpretations with results from other studies, particularly those conducted at the same site. This approach currently makes the narrative confusing and diminishes the perceived novelty of the study.

Thanks for your critical comments. We will focus more on the data we have, and reduce the part that related to previous studies as suggested. But the comparison is also important, regarded as one of the novelties of this study, because this is the one of the first field whole-soil warming studies that has been conducted for more than 10 years continuously, and focus on subsoils. This allows us to have a better understanding of time resolved warming effects on the SOC dynamics.

**Specific Comments**

- **L26:** "warmed plots" → "Warmed plots"

  We will change the text as suggested.

- **L47:** What does "p.2" mean? The same issue appears at **L67**.

  Thanks, we will fix this.

- **L75–76:** Why would "multiple enzymatic steps for depolymerization" lead to lower carbon use efficiency (CUE)? Please clarify the underlying mechanism.

  It is generally agreed that CUE is lower when microorganisms use complex polymers as substrates when compared to labile, and simple substrates. We will make this clearer and add a specific citations (Manzoni et al., 2012).

- **L89–91:** This sentence is unclear and should be rephrased for clarity.

  We will increase the clarity of this sentence as below:

  As subsoil stores half of the SOC in the top 1 m of soil (Scharlemann et al., 2014) and is regarded as slowly-cycling and large C reservoir (Harrison et al., 2011; Rumpel and Kögel-Knabner, 2011; Sierra et al., 2024). Therefore, it is imperative to understand how subsoil carbon will respond to future warming.

- L149: This statement contradicts the description in the Abstract (Line 27).

  We will fix the term in the Abstract.

- **L132–133, L155–156, L159:** The citation format makes these sentences difficult to understand. Please revise.

  We will revise according to the comment.

- **L164:** What does "replicates" mean here? Are these biological field replicates? Please clarify.

  Analytical replicates. We will make this clearer in the text.

- **L192:** I suggest adding soil pH values here for reference.

  Previous study has measured the pH values at the same study site (Rowley et al., 2025). The soils are acidic and have a pH ($H_2O$) of 4.9±0.1 (standard error of the mean), ranging between extremely and slightly acidic (3.7-6.2). We also measured the pH of our samples and will add the results in the SI.

- **L260:** The results presented in Table S1 are highly important for this study. Why are they not included in the main text? Additionally, the standard errors for some values (e.g., warmed soil at 20–30 cm depth) are relatively large. Please check data quality and consider whether outliers should be identified and addressed.

  For every sample, we have measured at least two analytical replicates, and for those with high standard deviation, we conducted more measurements to ensure data quality. For each run of measurement, we measured two different types of standard (Chernozem and Caffeine), and the standard deviations of both standards, and between each run are < 5 %. The larger errors could be derived from spatial heterogeneity between blocks in natural conditions.

- **L274:** According to Figure 1, fPOM at 40–50 cm depth appears to be higher under warming, which contradicts the statement made here.

  Well spotted, this should be fPOM from the 80-90 cm. We will modify this.

- **L293–294:** This sentence is difficult to understand and should be revised for clarity.

  We will improve the clarity of this sentence as such:

  The proportion of oPOM was not significantly affected by depth or treatment, exhibiting consistent values.

- **L407:** "> 30 under warming cm" → "> 30 cm under warming"

We will modify according to the suggestion.

- **L422–443:** This section mainly discusses results from other studies and includes extensive speculation that is not directly supported by measurements in the current study. These parts appear redundant and should be substantially reduced.

Thanks for the critical comments. We want to explore the mechanisms to explain why some of the changes have happened. But we sometimes don't have direct evidence to support our assumptions, therefore, we referred to similar experiments in the same ecosystem or at the same study site. But as mentioned before, we will refocus the Discussion section on our own statistically significant observations.

**References**

Manzoni, S., Taylor, P., Richter, A., Porporato, A., and Ågren, G. I.: Environmental and stoichiometric controls on microbial carbon-use efficiency in soils, New Phytologist, 196, 79–91, https://doi.org/10.1111/j.1469-8137.2012.04225.x, 2012.

Ofiti, N. O. E., Zosso, C. U., Soong, J. L., Solly, E. F., Torn, M. S., Wiesenberg, G. L. B., and Schmidt, M. W. I.: Warming promotes loss of subsoil carbon through accelerated degradation of plant-derived organic matter, Soil Biology and Biochemistry, 156, 108185, https://doi.org/10.1016/j.soilbio.2021.108185, 2021.

Riley, W. J., Tao, J., Mekonnen, Z. A., Grant, R. F., Brodie, E. L., Pegoraro, E., and Torn, M. S.: Experimental Soil Warming Impacts Soil Moisture and Plant Water Stress and Thereby Ecosystem Carbon Dynamics, Journal of Advances in Modeling Earth Systems, 17, e2024MS004714, https://doi.org/10.1029/2024MS004714, 2025.

Soong, J. L., Castanha, C., Hicks Pries, C. E., Ofiti, N., Porras, R. C., Riley, W. J., Schmidt, M. W. I., and Torn, M. S.: Five years of whole-soil warming led to loss of subsoil carbon stocks and increased $CO_2$ efflux, Sci. Adv., 7, eabd1343, https://doi.org/10.1126/sciadv.abd1343, 2021.

Zosso, C. U.: Are we losing it? Exploring subsoil organic carbon dynamics in a warming world on the molecular level, University of Zurich, 2022.

---

## Author Comment (AC2)

**RC2**

**General Comments**

This article reports the effects of ten years of experimental warming on soil carbon fractions at three depths from Blodgett Forest. Subsoil (80-90 cm depth) particulate organic matter (POM) mass was reduced and bulk soil carbon shifted composition toward more recalcitrant compounds (lignin and aromatic bonds), while mineral-associated organic matter (MAOM) remained unchanged, indicating that POM is more responsive to warming than MAOM.

The study has important implications for impacts of global change on soil carbon cycling, and represents a novel contribution to the field by reporting specific changes in carbon compound and organic matter composition with sustained warming.

However, there is a concerning over-interpretation of statistically non-significant results. The confidence intervals reveal massive uncertainty, which should preclude strong conclusions about subsoil carbon loss at specific depths. In addition, the reported marginally significant results are hard to justify within the confidence intervals that span a huge range of negative to positive values. At the very least, these parts of the results section should be reworded to reflect the high uncertainty instead of the supposedly "marginal significance." The statistical power of the comparisons is weak with n=3 samples throughout, and there were no reported corrections for the multiple comparisons made.

There is weak evidence for the bulk SOC composition shifts (Section 3.3, Fig. 3), which are based largely on visual PCA inspection. There are no formal statistical tests for whether warmed vs. control groups differ significantly in composition.

There are a few sentences in the results section that start with the word "basically," which detracts from the meaning of the sentence. Recommend removing or rephrasing.

Throughout the discussion section, my suggestion is to separate the pattern description from statistical inference.

The authors do a good job of putting the results from their study in the context of previous studies.

Throughout the manuscript, the authors explain topsoil patterns by inferring warming-induced increases in plant inputs (e.g., lines 377-384, 432-436, 485-487).

However, plant inputs were not measured in this study and the Ofiti et al. (2021) study they reference reported decreased root biomass at this site under warming, contradicting the proposed mechanism. Alternative explanations (moisture effects, substrate limitation, redistribution) receive insufficient consideration and the Abstract and key sections present this inference as established rather than hypothetical. I recommend revising to: (a) clearly identify increased inputs as a hypothesis rather than observation, (b) address the contradiction with Ofiti et al.'s root data, (c) discuss alternative mechanisms with appropriate weight, and (d) acknowledge this limitation explicitly. Consider tempering mechanistic conclusions in the Abstract and Conclusions to reflect this uncertainty.

Overall, the claim that MAOM remained stable under warming, across depths, is well-supported. The claim that POM is more responsive than MAOM to warming is somewhat supported, with a consistent pattern but mixed statistical significance. The claim that subsoil carbon loss is driven by POM depletion is overstated based on the evidence presented. While the bulk SOC interaction is significant, the individual subsoil depth reductions are not significant and the confidence intervals include substantial gains, not just losses. The claim that bulk SOC shifted toward lignin/aromatic bonds in warmed subsoils is also not well-supported, since it is based on visual interpretation of PCA only and Fig 3 shows trends but overlapping confidence ellipses.

We are deeply grateful for your thorough reading of the manuscript. Your critical and constructive feedback, and detailed, practical suggestions will significantly improve the quality of our manuscript. We will implement your suggestions accordingly.

We fully agree with your comments on wording in the results section, overestimating of certain statistical results, and will reword our interpretation of the potential increased plant inputs to the system from direct observations and instead highlight them as hypothesis. Accordingly, we will revise the manuscript based on your suggestions, find alternative explanations, and explicitly address the problems of uncertainties by small sample size, and the limitation of the study. Thanks for your comments.

We agree that adding statistical analysis on functional groups data would be beneficial. However, because DRIFTS data is only semi-quantitative, the reliability of statistical test would also be challenged, which is instead typically presented in principal

component analysis. However, to address your comment we will add a table into the supplementary information where we directly test the area under the curve values.

We also acknowledge that limited replication (n = 3 blocks) reduces statistical power, and results in our studies, meaning that marginal effects, should be interpreted with caution. However, large manipulation experiments such as the Blodgett Forest whole-soil warming experiment are typically limited in replication number. Because post hoc comparisons were only conducted following significant interaction terms and were restricted within depth increments, we did not apply additional multiple-comparison corrections, consistent with recommendations for hypothesis-driven mixed-effects modeling. We have chosen conservative statistical methods, which led to larger p-values compared to some other statistical methods to help support this.

**Specific Comments**

Were there changes in soil moisture with warming and by depth? This is important for the discussion of decomposition dynamics.

We will add this to discussion for example from Riley et al. (2025) and Soong et al. (2021). Briefly, warming significantly decreases soil moisture, and this reduction is most pronounced at topsoil (10-20 cm) and deep soil (80-90 cm; Pegoraro et al., 2024). The soil moisture at surface fluctuates magnificently between dry summer and moist winter (Soong et al., 2021).

L257-258: This could be more clearly worded to indicate that it is the interaction effect that is significant. Perhaps "Warming reduced SOC concentration in the subsoil but not the topsoil (warming × depth interaction: p = 0.002)."

We will simplify this sentence and other part of results accordingly.

L259-262: Reporting these statistics as marginal effects may be inaccurate, since the confidence interval is very large and the p-value close to non-significant. Recommend rephrasing to reflect the high uncertainty instead. Within that confidence interval, the effect could be anything from a huge loss to a moderate gain.

We will rephrase to highlight the uncertainties as follows:

At 60–70 and 80–90 cm, mean SOC concentrations were lower by 54% (CI: -83 %, 26 %) and 56% (-84 %, 20 %), respectively, but the wide confidence intervals and

marginally significant p-values ($p = 0.099$ and $p = 0.086$, respectively) indicate substantial variability rather than a definitive loss.

L271: Awkward/unclear wording "affected distinguishably"

We will clarify the wording as follows:

Depth significantly reduced fPOM concentrations ($p < 0.001$), while the impact of the treatment differed across depths ($p = 0.001$).

L271-273: Again, the large confidence intervals detract from the significance that is claimed. A non-significant increase could actually be a decrease, increase, or major increase, but the directionality of it is highly uncertain.

We will rephrase the sentence as follows:

At 10-20 and 40-50 cm, warming on average increased fPOM concentration respectively by 56 % and 97 %, but these increases were weak due to large standard errors.

L292-294: Unclear what is meant by "main effects" in this sentence.

We will simplify it by deleting „and main effects "

L314-318: "warmed subsoils displayed a trend towards increased AUC values" – this is descriptive, not statistically validated

We will now support this with statistical evidence in the supplementary information.

L334: Specify which PCA plot is being referenced.

We will implement the change.

L348-349: Reword for clarity.

We will reword as follows:

We did not find significant effects of warming, but significant effects of depth ($p < 0.001$; Fig. 5) on the bulk soil DRIFT stability index (DSI).

DSI analysis (Section 3.5) shows no significant warming effect for bulk soil despite claims.

We will change the title accordingly.

L355-361: Any mention of significance or non-significance should be accompanied by a p-value or p-value range.

We will add the p-value range to non-significance.

L382: If the relationship wasn't significant, it should not be interpreted as though it was. It is also unclear whether the non-significance is related to just one or more of those relationships listed.

We will rephrase and focus on the significant results.

L434: There is nothing in the code/data availability section. However, the authors are commended on their inclusion of significant amounts of data in the supplemental.

Thank you for your comment, we haven't added this yet. However, we can confirm that all data will be made available through the ESS-dive website as is customary for data published in collaboration with the Blodgett Forest whole-soil warming project.

**Technical Revisions**

L26: period instead of comma or else missing capitalization

L375: Significant not significantly

L396: Subsoil not subsoils

We will implement all the three suggestions.